

# Association of potential salivary biomarkers with diabetic retinopathy and its severity in type-2 diabetes mellitus: a proteomic analysis by mass spectrometry

Chin Soon Chee[1],[*], Khai Meng Chang[1],[*], Mun Fai Loke[2],
Voon Pei Angela Loo[1] and Visvaraja Subrayan[1]

[1] Department of Ophthalmology, University of Malaya, Kuala Lumpur, Malaysia
[2] Department of Medical Microbiology/Faculty of Medicine, University of Malaya, Kuala Lumpur, Malaysia
[*] These authors contributed equally to this work.

Corresponding authors
Mun Fai Loke, lmunfai@gmail.com
Voon Pei Angela Loo,
eyespecialist_loo@yahoo.com

## ABSTRACT

**Aim/hypothesis:** The aim of our study was to characterize the human salivary proteome and determine the changes in protein expression in two different stages of diabetic retinopathy with type-2 diabetes mellitus: (1) with non-proliferative diabetic retinopathy (NPDR) and (2) with proliferative diabetic retinopathy (PDR). Type-2 diabetes mellitus without diabetic retinopathy (XDR) was designated as control.
**Method:** In this study, 45 saliva samples were collected (15 samples from XDR control group, 15 samples from NPDR disease group and 15 samples from PDR disease group). Salivary proteins were extracted, reduced, alkylated, trypsin digested and labeled with an isobaric tag for relative and absolute quantitation (iTRAQ) before being analyzed by an Orbitrap fusion tribrid mass spectrometer. Protein annotation, fold change calculation and statistical analysis were interrogated by Proteome Discoverer. Biological pathway analysis was performed by Ingenuity Pathway Analysis. Data are available via ProteomeXchange with identifiers PXD003723–PX003725.
**Results:** A total of 315 proteins were identified from the salivary proteome and 119 proteins were found to be differentially expressed. The differentially expressed proteins from the NPDR disease group and the PDR disease group were assigned to respective canonical pathways indicating increased Liver X receptor/Retinoid X receptor (LXR/RXR) activation, Farnesoid X receptor/Retinoid X receptor (FXR/RXR) activation, acute phase response signaling, sucrose degradation V and regulation of actin-based motility by Rho in the PDR disease group compared to the NPDR disease group.
**Conclusions/Interpretation:** Progression from non-proliferative to proliferative retinopathy in type-2 diabetic patients is a complex multi-mechanism and systemic process. Furthermore, saliva was shown to be a feasible alternative sample source for diabetic retinopathy biomarkers.

## INTRODUCTION

Early onset of type-2 diabetes mellitus has been devastating and a major epidemic across the world. Report shows that 7% of newly diagnosed type-2 diabetic patients in the U.S. have been diabetic for approximately 4–7 years before diagnosis (*Rao et al., 2009*). There is a very low global awareness and precaution on how to prevent type-2 diabetes. Patients with prolonged type-2 diabetes without proper consultation and medication have a higher probability of developing complications such as diabetic retinopathy, which can eventually lead to blindness.

Diabetic retinopathy is one of the most common and severe microvascular complications of type-2 diabetes. Symptoms of diabetic retinopathy were retinal ischemia and increased retinal vascular permeability which leads to vision loss or blindness ultimately. Diabetic retinopathy could be classified into two main stages: non-proliferative diabetic retinopathy (NPDR) and proliferative diabetic retinopathy (PDR) in terms of the severity. The NPDR patients had lesions on the eye vasculature layer and vision would be lost if there was fluid in the central portion of the eyes (*Csosz et al., 2012*). PDR was literally characterized by pathological retinal vascular leakage (macular edema) and retinal neovascularization (*Gao et al., 2008*). There are several studies reported on the factors related to pathogenesis of PDR (*Tarr et al., 2013*), e.g., vascular endothelial growth factor for the proliferation and propagation of blood vessels in eyes, angiotensin-converting enzyme, insulin-like growth factor, angiopoietin, erythropoietin, placenta growth factor, advanced glycation end product, and antiangiogenic factors like pigment epithelium-derived factor.

The noninvasive nature and simple collection allows repetition and multiple collection of saliva that can potentially aid in early diagnosis, monitoring disease progression, or treatment responses with minimally trained personnel. This advantage of using saliva attracts investigators who are looking for an alternative body fluid to simplify diagnostic procedures (*Giusti et al., 2007a*; *Giusti et al., 2007b*; *Hu, Loo & Wong, 2007a*; *Peluso et al., 2007*). Secretions from salivary glands, oral mucosa, periodontium, and oral microbiota all contribute to the final content of saliva. Saliva, a complex balance from local and systemic sources, allows for applications in the diagnosis not only for salivary gland disorders but also for oral diseases and systemic conditions (*Caporossi, Santoro & Papaleo, 2010*; *Good et al., 2007*; *Hu et al., 2007b*; *Lee, Garon & Wong, 2009*). In our study, saliva samples from XDR, NPDR and PDR patient groups are selected as a diagnostic fluid to study the salivary proteome. Saliva offers several advantages over vitreous humor, tears and serum. The availability of saliva make it the simplest and non-invasive way of body fluid collection allowing repetitive collection. Saliva collection decreases the risk of contracting infectious diseases compared to other body fluids collection and it is convenient to obtain saliva from children or handicapped patients, in whom blood sampling could be inconvenient. Saliva is an ideal body fluid for the purpose of biomarker identification based on several decent studies. *Castagnola et al. (2011)* reported that there was existence of specific salivary biomarkers associated with a health or disease state. In the meanwhile, *Shinkai et al. (2004)* also reported that there was an altered saliva

composition in type-2 diabetic patients. Hence, it is important to investigate the salivary proteome profiles for diabetic retinopathy complications. Moreover, biomarkers from salivary proteome of diabetic retinopathy are yet to be discovered.

In recent years, advancement in proteomic technology has invented plenty of instruments for proteomics research. A sophisticated mass spectrometer, the Orbitrap fusion tribrid mass spectrometer is used in our study to achieve our objective. The Orbitrap fusion tribrid mass spectrometer, comprises of a mass filter, a collision cell, a high-field Oribitrap analyzer and a dual cell linear ion trap analyzer, offers high $MS^2$ acquisition speed of 20 Hz (*Senko et al., 2013*). We believe that this new system, with its fast scan rate, could provide more comprehensive proteome analysis within shorter time. The development of higher energy collision dissociation (HCD) in the LTQ-Orbitrap has also overcome the 1/3 rule limitation that restricts the analysis of product ions with m/z values less than 25–30% of the precursor ion in traditional ion-trap instruments (*Rauniyar & Yates, 2014*). Limitations with analyzing biological samples of complex nature, such as the salivary proteome, are the masking of low-abundance proteins by the preponderance of a small number of highly abundant salivary proteins and the high dynamic range of such proteome that precludes the use of conventional proteomic strategies (*Hu et al., 2005*; *Hu, Loo & Wong, 2007a*). A method that has been proposed to largely overcome these deficits is isobaric labeling isobaric tags for relative and absolute quantitation (iTRAQ) (*Casado-Vela et al., 2010*; *Rauniyar & Yates, 2014*). These isotope tags permit ready discrimination by mass spectrometry (MS), thereby permitting comparative quantification to a reference sample in a multiplex manner and the examination of different samples in a single mass spectrometric analysis with good quantification precision. Hence, the ratio cutoff applied for significant protein change via the iTRAQ approach is lower than the cutoff applied for the label-free quantification approach (*Rauniyar & Yates, 2014*). This is the first attempt to analyze the salivary proteome profiles of type-2 diabetes complicated with diabetic retinopathy using the high resolution and accurate mass Orbitrap fusion tribrid mass spectrometer.

## METHODOLOGY

### Sample collection and processing

Saliva sample from 45 subjects with type-2 diabetes mellitus were collected. Subjects for this study were recruited from patients who visited the Eye clinic at the University of Malaya Medical Center (UMMC) during the period between November 2013 and April 2014. Patients older than 45 years old diagnosed with type-2 diabetes for more than five years with or without diabetic retinopathy were included in this study. All the patients were on oral medication for glycemic index control and/or dyslipidemia (none of them were on insulin therapy). The following patients were excluded from the study: (1) patients who had oral surgery or treatment within the past three months; (2) patients who had active gum bleeding; (3) patients with dry mouth (e.g. Sjögren's syndrome); (4) patients who had recent oral injury; (5) patients with history of malignancy, autoimmune diseases, Hepatitis/HIV infection; (6) patients on any types of eye drops for active eye disease (e.g. glaucoma, conjunctivitis); (7) patients who had significant

ocular medium opacities such as cataract or hazy cornea; (8) patients who had intravitreal injection and/or retinal laser treatment prior to diagnose for diabetic retinopathy; (9) patients with quiescent PDR and (10) smokers. Patients were classified by their severity of diabetic retinopathy according to the International Clinical Classification System for Diabetic Retinopathy and Diabetic Macular Edema by American Academy of Ophthalmology (*Wilkinson et al., 2003*). Diabetic retinopathy was graded through clinical fundus examination photography by two independent eye specialists. Subjects were grouped into three groups based on their clinical presentation: (1) type-2 diabetes mellitus without diabetic retinopathy (XDR) as control, (2) type-2 diabetes with NPDR and (3) type-2 diabetes with PDR (Table 1). PDR patients with active neovascularization were included.

Subjects fasted overnight for at least 8 h (except for drinking) prior to the collection of saliva samples. They were instructed to avoid drinks containing caffeine and alcohol for 12 h and avoid vigorous physical activity for 4 h prior to sample collection. In addition, they were also reminded to avoid brushing teeth 1 h prior to sample collection and avoid applying lipstick. Saliva samples were collected between 9–10 a.m. The subjects were asked to rinse their mouths thoroughly with sterile water 10 min before sample collection, then to tilt their heads forward and allow saliva to flow into a sterile centrifuge tube until 5 mL of saliva was collected. Saliva samples were spun at 8,000 × g for 20 min at 4 °C to spin down nuclei, cell debris and bacteria cells. The supernatant was then kept at −20 °C for subsequent analysis.

This study was approved by the Medical Ethics Committee of UMMC (Reference number: 1017.28) and written informed consent was obtained from the patients prior to samples collection.

## Proteins extraction

Salivary protein was extracted by acetone precipitation method as described by *Vitorino et al. (2012)* with modification. Saliva samples were precipitated by mixing with six volumes of pre-chilled acetone (Grade AR) (Friedemann Schmidt, Parkwood, Perth, Australia) and mixed by vortexing. Each sample was allowed to stand overnight at 4 °C. After incubation, all samples were centrifuged at 12,000 × g for 30 min. The supernatant and pellet were separated. The pellet was dried at room temperature.

## Protein concentration

Protein concentration was determined using Bradford assay (Bio-Rad, Hercules, California, USA) with bovine serum albumin (BSA) as standard (*Bradford, 1976*). Protein standards and tests were prepared in triplicate.

## Reduction, alkylation and trypsin digestion of salivary proteins

Reduction, alkylation and trypsin digestion of salivary proteins were carried out according to the method described by *Ross et al. (2004)* with modification. Briefly, 50 μg of salivary protein was suspended in 100 mmol/l triethylammonium bicarbonate (pH 8.5) (Sigma-Aldrich, St. Louis, Missouri, USA) and vortex to make sure the pellet was completely

**Table 1 Demographic of subjects.**

| Parameters | XDR (N = 15) | NPDR (N = 15) | PDR (N = 15) |
|---|---|---|---|
| Age | 61.8 ± 8.77 | 60.63 ± 6.49 | 58.94 ± 6.98 |
| Race (M/I/C) | 7/3/5 | 7/5/3 | 9/4/2 |
| Sex (Male/F) | 5/10 | 7/8 | 8/7 |
| Duration of diabetes (year) | 12.87 ± 4.97 | 13.94 ± 7.15 | 14.62 ± 5.51 |
| HbA$_{1c}$ (%) | 7.73 ± 1.15 | 8.43 ± 1.08 | 8.85 ± 1.9 |
| Fasting blood sugar (mmol/l) | 8.16 ± 1.62 | 8.6 ± 3.37 | 8.99 ± 3.3 |
| Creatinine (µg/l) | 93.9 ± 41.17 | 107.0 ± 40.9 | 125.3 ± 71.86 |

**Notes:**
M, Malays; I, Indian; C, Chinese; F, Female.
All the pairs were compared using one-way ANOVA and Student's t-test, there no statistically significant difference (p-value all > 0.05).

dissolved. Protein reduction was carried out by adding 10 mmol/l tris-(2-carboxyethyl)-phosphine (Sigma-Aldrich, St. Louis, Missouri, USA) and incubated at 60 °C for 60 min. Reduced protein was subsequently alkylated with 20 mmol/l iodoacetamide (Bio-Rad, Hercules, California, USA) in the dark for 60 min at room temperature. Finally, the protein samples were digested with 1 µg of MS grade porcine trypsin (Calbiochem, La Jolla, California, USA) at 37 °C for 16–18 h. The reaction was terminated by adding trifluoroacetic acid (Sigma-Aldrich, St. Louis, Missouri, USA) to the final concentration of 5% (v/v).

## iTRAQ labeling of salivary peptides

Digested peptides samples were labeled using the iTRAQ 8Plex Multiplexing kit (AB Sciex, Foster City, California, USA) according to the manufacturer's protocol. Peptides from XDR, NPDR and PDR patient groups were labeled with isobaric tags 113–115 respectively at room temperature for 4 h. The reaction was quenched with 20 mmol/l Tris (pH 8.0) (Sigma-Aldrich, St. Louis, Missouri, USA). The contents of each iTRAQ reagent labeled sample tubes were combined.

## Peptide purification and concentration

Pierce C18 Spin Column (Thermo Scientific, Rockford, Illinois, USA) was used to purify and concentrate the labeled peptides according to the manufacturer's protocol.

## Liquid chromatography-mass spectrometer (LC-MS) analysis

Ten micrograms of salivary digest were separated on the EASY-nLC 1000 (Thermo Scientific, San Jose, California, USA) using the Acclaim PepMap C$_{18}$ (3 µm, 75 µm × 50 cm) column (Thermo Scientific, San Jose, California, USA). Solvent A was HPLC-grade water with 0.1% (v/v) formic acid, and solvent B was HPLC grade acetonitrile with 0.1% (v/v) formic acid. Separation was performed with stepwise gradient (5–30% B for 185 min, 30–50% B for 20 min, 50–95% B for 20 min) at 300 nl/min over 225 min. MS data was generated using an Orbitrap fusion tribrid mass spectrometer (Thermo Scientific, San Jose, California, USA) operated with −2.5 kV (positive ions) applied to the central electrode. The mixture of isotopolog peptides were analyzed by

combining scan events from two Single ion monitoring (SIM) modes. The first full time scan mode MS employed a scan range (m/z) of 380–2,000, Orbitrap resolution of 120,000, target automatic gain control (AGC) values of 200,000 and a maximum injection time of 50 ms. The second scan mode, HCD-MS/MS was performed at the Quadrupole with the isolation width of 1.6 Th, HCD fragmentation with normalized collision energy of 35%, Orbitrap resolution of 30,000, target AGC values of 50,000, and a maximum injection time of 60 ms. Only precursors with charge state 2–7 were subjected to $MS^2$. Monoisotopic precursor selection and dynamic exclusion (70 s duration, 10 ppm mass tolerance) were enabled. Analysis was carried out with 3 technical replications.

## Data analysis

The raw data was processed using Proteome Discoverer version 1.4 (Thermo Scientific, San Jose, California, USA). MS/MS spectra were searched with Sequest engine against *Homo sapiens* database using the following parameters: full trypsin digest with maximum 2 missed cleavages, fixed modification carbamidomethylation of cysteine (+57.021 Da), variable modification oxidation of methionine (+15.995 Da) and iTRAQ 8-plex modification of lysine and peptide N termini (+304.205 Da). Precursor mass tolerance was 10 ppm and product ions fragment ion tolerance was 0.02 Da. Peptide spectral matches were validated using percolator based on q-values at a 1% false discovery rate. iTRAQ ratio reporting was pair wise: NPDR/XDR (114/113) and PDR/XDR (115/113).

## Bioinformatic analysis of differential expressed proteins

Differentially expressed proteins from NPDR and PDR patient groups were further analyzed using Ingenuity Pathway Analysis (IPA) (version 8.8) (Qiagen, Redwood, California, USA) to statistically determine the functions and pathways associated with each of the individual proteins. Accession number for each of the proteins and the fold change between NPDR and PDR groups relative to XDR group were tabulated. IPA utilized the Ingenuity Pathways Analysis Knowledge Base (IPA KB), a manually curated database of protein interactions from the literature, for analysis. A fold change cut-off of 1.5 was set to identify significant differentially regulated proteins. A list of networks and functional and canonical pathways were generated and the significance of the associations was assessed with the Fisher's exact test ($p < 0.05$).

The MS proteomics data have been deposited to the ProteomeXchange Consortium via the PRIDE (*Vizcaíno et al., 2016*) partner repository with the dataset identifiers PXD003723–PXD003725.

## RESULTS

Based on the criteria that at least one unique peptide and a minimum of two peptides match for protein identification, 315 proteins could be identified from the salivary proteome. The mean percentage of peptide coverage was 35.17% ± 2.55 ranging from 1.72–87.67%. The overall salivary proteome was annotated using GO annotation (GO) analysis facilitated by Proteome Discoverer version 1.4 and ProteinCenter database. Salivary proteins were assigned according to three different classifications: cellular

component classification, biological process classification and molecular function classification. Of which, 19% were cytoplasmic proteins, 19% were extracellular proteins, 12% were membrane proteins and 11% were proteins localized in the nucleus (Fig. S1A). Metabolic proteins comprised 15% of the proteins identified, 13% were involved in regulation of biological process and 12% were proteins that respond to stimulus (Fig. S1B). As high as 29% of the proteins were involved in protein binding, 18% showed catalytic activities and 11% was involved in metal ion binding (Fig. S1C).

For quantitative analysis, only proteins with complete labeled peptides were considered. iTRAQ data was expressed in pair ratio: NPDR vs XDR (iTRAQ 114/iTRAQ 113) and PDR vs. XDR (iTRAQ 115/iTRAQ 113). Only those with fold-change < 0.5 or > 2 were considered to be differentially expressed. A total of 119 proteins were found to be differentially expressed. Figure 1 illustrates the comparison of the log ratio of the relative intensity (NPDR/XDR; PDR/XDR) for proteins commonly found in XDR, NPDR and PDR disease groups. Figure 2 presents the comparison of the log ratio of the relative intensity (NPDR/XDR; PDR/XDR) for proteins unique to XDR and NPDR or PDR disease groups. Among those that are differentially expressed, 1 protein was un-regulated in NPDR and PDR compared to XDR disease groups. Eighty-two proteins were increased in PDR compared to XDR disease groups but decreased in NPDR in comparison to XDR disease groups. Two proteins were down-regulated in NPDR compared to XDR disease groups but not detected in PDR disease group. The remaining 34 proteins were increased in PDR relative to XDR disease groups but not found in NPDR disease group.

A total of 117 salivary proteins were increased in PDR disease group relative to XDR disease group. Eighty-two increased salivary proteins in PDR disease group were decreased in NPDR disease group, 34 were not found in NPDR disease group and metalloproteinase inhibitor 1 precursor was increased in both PDR and NPDR disease groups. Table S1 lists the top 26 most up-regulated salivary proteins with a minimum fold change of 20 by relative protein abundance. Among the top 20 proteins that were increased in PDR disease group, 13% were predicted to respond to stimulus, 10% were predicted to regulate biological process, 9% were involved in metabolism, 8% were involved in cell organization and biogenesis and 8% were predicted to be involved in defense response (Fig. S2B). Most of these proteins were predicted to have protein binding capability (28%) and 17% might have catalytic activity (Fig. S2C). On the other hand, peroxiredoxin-1 and unconventional myosin-IXb isoform 2 were decreased in NPDR disease group (Table S2).

Twenty-one interacting proteins and 1 highly increased protein (clusterin from NPDR disease group), together with 35 interacting proteins and 1 highly increased protein (tropomyosin alpha-3 chain isoform 2 from PDR disease group) (Table S3) generated 3 protein-protein networks (Fig. S3). Network (A) includes the diseases and functions of connective tissue disorders, immunological disease and inflammatory disease. Network (B) includes the diseases and functions of cellular movement, hematological system development and function and immune cell trafficking while Network (C) includes the diseases and functions of cellular growth and proliferation, cancer and carbohydrate

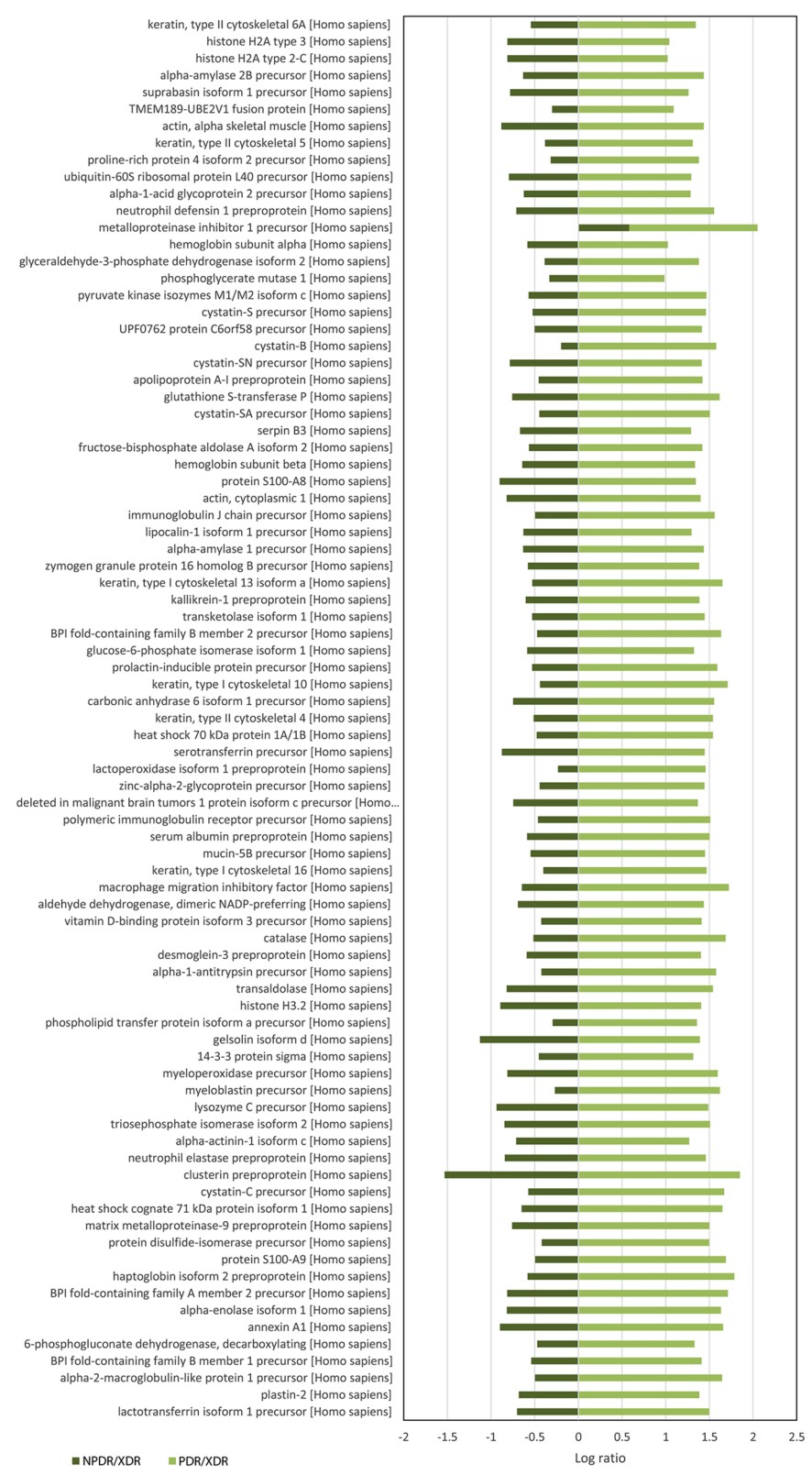

**Figure 1** Log ratio of relative intensity (NPDR/XDR; PDR/XDR) for proteins commonly found in XDR, NPDR and PDR disease groups.

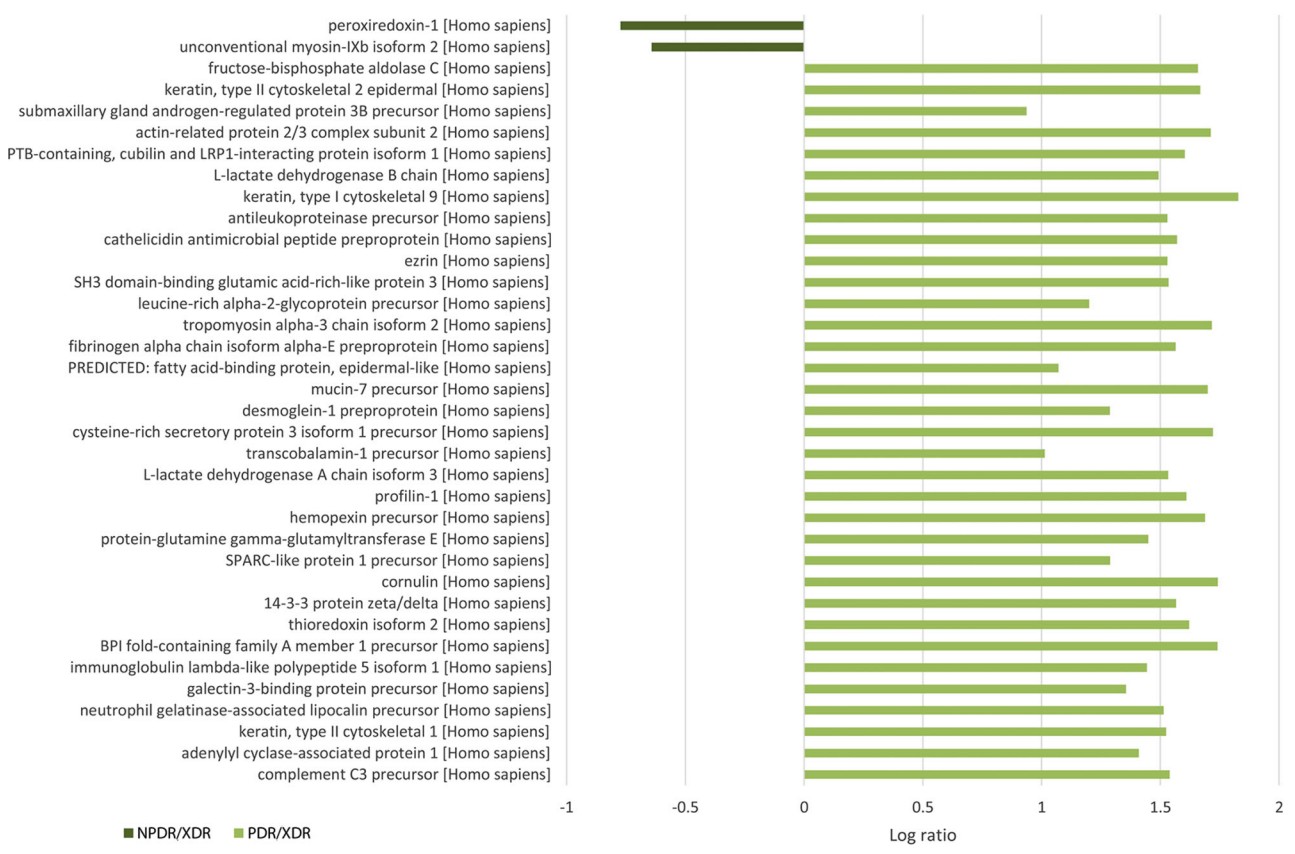

**Figure 2 Log ratio of relative intensity (NPDR/XDR; PDR/XDR) or proteins found exclusively in NPDR and PDR disease groups.**

metabolism. The top three canonical pathways with the highest–log (p-value) for NPDR disease group were Liver X receptor/Retinoid X receptor (LXR/RXR) activation, glycolysis I and clathrin-mediated endocytosis signaling while for PDR group the pathways were LXR/RXR activation, glycolysis I and Farnesoid X receptor/Retinoid X receptor (FXR/RXR) activation. Comparison between the differentially expressed proteins from NPDR and PDR disease groups in different canonical pathways indicated increased LXR/RXR activation, FXR/RXR activation, acute phase response signaling, sucrose degradation V and regulation of actin-based motility by Rho in PDR disease group compared to NPDR disease group (Fig. 3).

## DISCUSSION

To the best of our knowledge, the salivary proteome of diabetic retinopathy has not yet been characterized. Among top increased proteins in the PDR group, 8% were predicted to be defense proteins and 9% were metabolic proteins suggesting that the expression of salivary defense and metabolic proteins is related to diabetic retinopathy. This is consistent with the report by *Fernández-Real & Pickup (2008)* that defense response proteins were elevated in type-2 diabetic patients and this gradually led to surging of metabolic proteins. Most of these defense response proteins that were increased in PDR are involved with pro-inflammatory immune response and many had previously been



**Figure 3 Canonical pathways comparative studies in differentially expressed proteins from NPDR and PDR groups.**

reported to be associated with diabetes mellitus. Such defense response proteins include S100-A9 (*Cabras et al., 2010*), alpha-2-macroglobulin-like protein 1 (*James et al., 1980*), neutrophil elastase (*Collier et al., 1989*), alpha-1-antitrypsin (SERPINA1) (*Kalis et al., 2010*), cystatin-C (*Reutens et al., 2013*) and macrophage migration inhibitory factor (MIF) (*Tashimo et al., 2004*). Heterodimer of S100-A9 and S100-A8 known as myeloid-related protein-8/14 (MRP8/14) binds to receptor for advanced glycation end-products and Toll-like receptor-4 (TLR-4) thereby initiating the intracellular inflammatory signaling cascade (*Caseiro et al., 2013*). Alpha 2-macroglobulin has been suggested to be a potential biomarker for diabetic retinopathy and other diabetic complications (*Lu et al., 2013*). Neutrophil elastase was reported to be a marker for the development of diabetic angiopathy (*Piwowar, Knapik-Kordecka & Warwas, 2000*). Neutrophil releases neutrophil elastase that enhances inflammatory responses. SERPINA1 can suppress apoptosis of pancreatic *β*-cells that promote insulin secretion (*Kalis et al., 2010*). A novel immune system regulatory pathway involving SERPINA1 and complement 3 (C3) was unveiled recently (*Sahu & Lambris, 2001*). Activation of C3 promotes phagocytosis, supports local inflammatory responses against pathogens and initiates the humoral immune response; on the other hand, its activation leads to host cell damage. However, binding of SERPINA1 to C3 inhibits and regulates the cleavage and activation of C3 during inflammation. MIF produce by T lymphocytes and macrophage can initiate local inflammation through the inhibition of the random movement of macrophage and enhance their adhesion (*Mitamura et al., 2000*). Capillary occlusion can be caused by the natural tendency of leucocytes and macrophages to adhere to endothelium that eventually results in retinal ischemia seen in diabetic retinopathy (*Schröder, Palinski & Schmid-Schönbein, 1991*). MIF may play a role during the proliferative phase of diabetic retinopathy by activating and retaining intraocular macrophage. Furthermore, MIF interacts with peroxiredoxin-1 by reducing tautomerase and oxidoreductase activities of MIF and inhibits the activity of peroxiredoxin-1 (*Kudrin & Ray, 2008*). Our result show that peroxiredoxin-1 was decreased in NPDR and not found in PDR that was in line with the result reported by *Rao et al. (2009)*.

The up-regulation of BPI fold-containing family A member 1/2, BPI fold-containing family B member 2 and neutrophil gelatinase-associated lipocalin in the PDR patient group suggested that innate immune response might also be involved in PDR. This might also suggest the involvement of microbial agents in PDR pathogenesis. Binding of BPI to endotoxin of Gram-negative bacteria outer membrane could trigger sub-lethal and lethal effects on the bacteria and neutralize the activity of endotoxin (*Schultz et al., 2007*). Myeloperoxidase (MPO) and lactotransferrin isoform 1 (LTF) are abundantly expressed in neutrophil granulocytes with antioxidant, anticarcinogenic, antibacterial effects, implying an important role in innate immunity. During the oxidative burst of activated neutrophils, MPO utilize hydrogen peroxide and chloride anion to generate a highly reactive and cytotoxic product, hydrochlorous acid which are used by bactericidal (*Mütze et al. 2003*). Protein-protein networks analysis revealed that cellular target of LTF is MPO, to which LTF bind and inhibit MPO. Neutrophil gelatinase-associated lipocalin was demonstrated to be an early biomarker for diabetic nephropathy

(*Bolignano et al., 2009*). In addition, neutrophil gelatinase-associated lipocalin is an iron-binding protein that may inhibit the growth of bacteria by depleting the iron source of bacteria (*Cherayil, 2011*).

In addition, neutrophil gelatinase-associated lipocalin can also activate pro matrix metalloproteinase-9 (MMP-9) (*Tschesche et al., 2001*). Hyperglycemia-induced activation of MMP-9 promotes apoptosis of retinal capillary cells and can result in development of diabetic retinopathy (*Kowluru, 2010*). Interestingly, metalloproteinase inhibitor 1, an inhibitor of MMP-9, was found to be increased in both NPDR and PDR patients. In contrast, MMP-9 was low in XDR patients. *Florys et al. (2006)* reported that high blood glucose concentration could induce the expression of metalloproteinase inhibitor 1. Thus, our results suggest that metalloproteinase inhibitor 1 may influence the development of diabetic retinopathy and combined with high levels of MMP-9 may drive the progression towards the proliferative phase.

The retina is rich in unsaturated fatty acid, rapid oxygen uptake and glucose oxidation rate compared to other areas of the human body that renders the retina highly susceptible to oxidative stress. Heme is highly toxic due to its ability to cause protein aggregation and produce lipid peroxide from lipid peroxidation that could contribute to oxidative stress. Hemopexin functions as a scavenger of heme. The finding of high level of hemopexin in the saliva of PDR patients supported the hypothesis that hyperglycemia, changes in the redox homeostasis and oxidative stress are key pathogenic events in diabetic retinopathy (*Kowluru & Chan, 2007*). Glycation end-products (AGEs) are produced by non-enzymatic glycation reactions of amino groups, lipids and DNA with glucose and its formation is an important pathogenic mechanism in diabetic retinopathy. AGEs have been linked to the breakdown of the inner blood retina barrier (iBRB) during diabetic retinopathy by modulating the expression of vasopermeability factor (*Amin et al., 1997*). Galectin-3-binding protein, an AGE-binding protein, can enhance the iBRB dysfunction in diabetes and play a significant role in AGE-related pathophysiology during diabetic retinopathy (*Pugliese et al., 2000*). Galectin-3-binding protein was also presented in relatively high abundance in PDR patients. High abundance of clusterin has been reported in vitreous humor of PDR patients (*Gao et al., 2008*). Thus, it is not surprising that our data also shows an unprecedented high abundance of clusterin in the saliva of PDR patients. Clusterin is believed to promote angiogenesis or vascular permeability, which contributes to the pathogenesis of diabetic retinopathy (*Wang et al., 2013*).

Glyceraldehyde-3-phosphate dehydrogenase (GAPDH), alpha-enolase isoform 1 (ENO1) and pyruvate kinase isozymes M1/M2 isoform c (PKM) are typical enzymes found in saliva that are involved in glycolysis and gluconeogenesis. Although GAPDH is a glycolytic enzyme, it has also been proven to have multiple cytoplasmic, membrane, and nuclear functions. *Saunders, Chalecka-Franaszek & Chuang (1997)* reported that GAPDH was a major intracellular messenger mediating apoptosis of cells and GAPDH translocation to the nucleus was considered a crucial step in glucose-induced apoptosis of retinal Muller cells. Moreover, the role of GADPH in the development and progression of diabetic retinopathy has been investigated by *Kanwar & Kowluru (2009)*.

**Table 2 Proteins associated with NPDR and PDR that were reported in vitreous.**

| Protein name | Saliva | Vitreous |
|---|---|---|
| Alpha-1-antitrypsin (SERPINA1) | Elevated in PDR | Elevated in moderate and severe PDR (*Gao et al., 2008*; *Kanwar & Kowluru, 2009*; *Hazra et al., 2012*) |
| Alpha-2-macroglobulin | Elevated in PDR | Elevated in severe PDR (*Gao et al., 2008*; *Kanwar & Kowluru, 2009*; *Hazra et al., 2012*) |
| Alpha-enolase | Elevated in PDR | Present in control and moderate PDR (*Kanwar & Kowluru, 2009*; *Hazra et al., 2012*) |
| Apolipoprotein A-I | Elevated in PDR | Elevated in moderate and severe PDR (*Gao et al., 2008*; *Kanwar & Kowluru, 2009*; *Hazra et al., 2012*) |
| Catalase | Elevated in PDR | Present in XDR and PDR (*Gao et al., 2008*; *Kanwar & Kowluru, 2009*) |
| Clusterin | Elevated in PDR | Present in vitreous (esp. moderate and severe PDR (*Gao et al., 2008*; *Hazra et al., 2012*); decreased in PDR (*Kadoglou et al., 2005*) |
| Complement C3 | Elevated in PDR | Elevated in moderate PDR (*Gao et al., 2008*; *Hazra et al., 2012*) |
| Cystatin-C | Elevated in PDR | Present in vitreous (control, moderate and severe PDR) (*Gao et al., 2008*; *Hazra et al., 2012*) |
| Fructose-bisphosphate aldolase C | Elevated in PDR | Present in XDR (*Gao et al., 2008*) |
| Galectin-3-binding protein | Elevated in PDR | Elevated in severe PDR (*Gao et al., 2008*; *Hazra et al., 2012*) |
| Gelsolin | Elevated in PDR | Present in moderate and severe PDR (*Hazra et al., 2012*) |
| Glyceraldehyde-3-phosphate dehydrogenase | Elevated in PDR | Decreased in PDR (*Kadoglou et al., 2005*); present in control and moderate PDR (*Hazra et al., 2012*) |
| Haptoglobin | Elevated in PDR | Present in vitreous (esp. severe PDR) (*Gao et al., 2008*; *Kanwar & Kowluru, 2009*; *Hazra et al., 2012*) |
| Hemoglobulin subunit alpha | Elevated in PDR | Elevated in PDR (*Gao et al., 2008*) |
| Hemopexin | Elevated in PDR | Elevated in XDR (*Gao et al., 2008*; *Kadoglou et al., 2005*),32]; present in control, moderate and severe PDR (*Kanwar & Kowluru, 2009*; *Hazra et al., 2012*) |
| Peroxiredoxin-1 | Decreased in NPDR | Elevated in PDR (*Gao et al., 2008*); present in control (*Hazra et al., 2012*) |
| Protein S100-A8 | Elevated in PDR | Present in XDR and PDR (*Gao et al., 2008*) |
| Protein S100-A9 | Elevated in PDR | Present in XDR and PDR (*Gao et al., 2008*) |

LXR/RXR activation, FXR/RXR activation, clathrin-mediated endocytosis signaling, acute phase response signaling and regulation of actin-based motility by Rho are highly associated with the pathogenesis and progression of diabetic retinopathy. Activation of LXR promotes reverse cholesterol transport and suppressed inflammatory response which in turn improve and inhibit the progression of diabetic retinopathy (*Hazra et al., 2012*). RXR is known to be associated with the progression of diabetic retinopathy (*Roy et al., 2009*), with RXR activation playing a key role in inhibiting high-glucose-induced oxidative stress, systemic lipid and glucose metabolism, energy homeostasis, and inflammatory control. The role of the farnesoid X receptor (FXR) in relation to diabetic retinopathy had not been reported at this time; however, the role of FXR in diabetic nephropathy (*Wang et al., 2010*) and atherosclerotic lesion formation (*Hartman et al., 2009*) are well established. FXR is involved in microvascular or macrovascular complication of diabetes; hence, FXR may be related to the pathogenesis of diabetic retinopathy. Clathrin-mediated endocytosis is involved in the internalization of the ligand-receptor complex through clathrin-coated vesicles that initialize the intracellular signal transduction cascade in response to the stimulus. AGEs are known to accumulate within the neural retina of diabetics but the effect on neural dysfunction and depletion

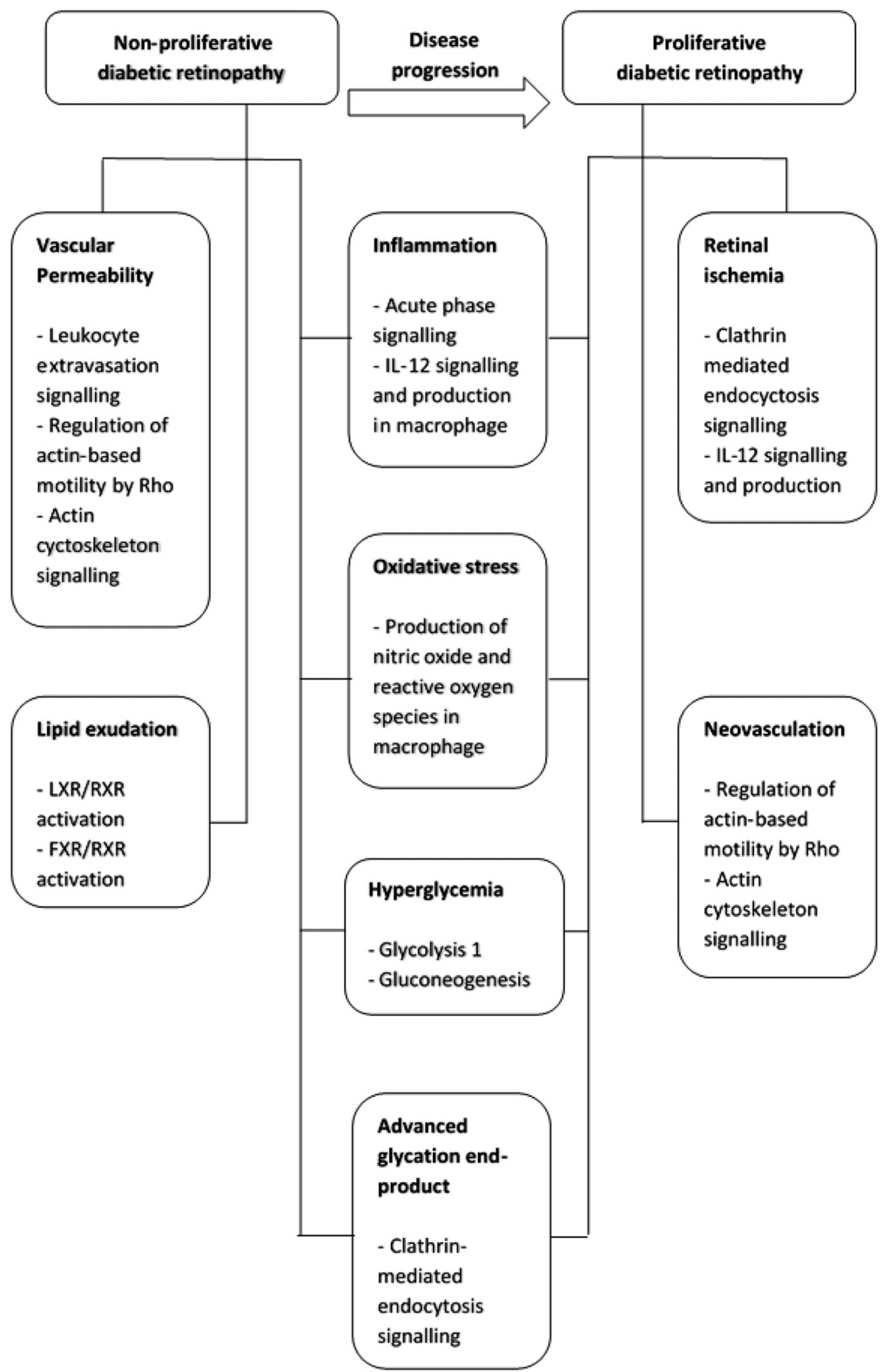

**Figure 4 Summary of pathogenetic mechanism for non-proliferative diabetic retinopathy and proliferative diabetic retinopathy and the functional pathways involved.**

during retinopathy was poorly investigated (*Stitt, 2003*). Retinal microvascular endothelial cells express AGE-receptor and mediate endocytic uptake of AGEs eventually leads to increased retinal vascular cells toxicity, affecting capillary function (*Stitt, 2003*). *Mizuno et al. (2010)* reported that cellular degeneration, remodeling and cell death leading to emerging of new blood vessels which was observed in PDR was the consequence of excessive glutamate up-take by retinal vascular endothelial cells. As expected, acute phase response signaling pathway had played a causative role in the pathogenesis of diabetic retinopathy. Acute phase response is generally considered an adaptive response that restores homeostasis. However, excessive or persistent overexpression of acute-phase proteins can lead to tissue and organ damage (*Gerhardinger et al., 2005*). GTPases of the Rho family regulate the interaction between cells and extracellular matrix resulting in angiogenesis, vascular permeability, leukocyte migration and platelet formation in vivo. In the early stage of angiogenesis, GTPase Rho facilitates the endothelial cell retraction and release of junctional complex simultaneously further facilitating the vascular leakage (*Cheresh & Stupack, 2008*). Neovasculation is the main event in the proliferative stage of diabetic retinopathy and GTPase Rho may be a key regulator enzyme in the early stage of angiogenesis. Comparing this profile of vitreous (*Gao et al., 2008*; *Wang et al., 2013*; *Yamane et al., 2003*; *Yu et al., 2008*), similarities were noted in Table 2. This demonstrates that local (vitreous) changes in protein levels associated with pathogenesis and progression of diabetic retinopathy may be reflected systemically in the saliva.

As other microvascular complications of diabetes also progress with inflammatory processes, serum creatinine was measured (Table 1) to exclude patients with severe diabetic nephropathy. However, patients with non-detectable microvascular complications were not excluded, which is a limitation of our study. Furthermore, although patients with detectable poor oral hygiene were excluded, it is not possible to rule out patients with mild salivary gland inflammation.

In conclusion, the progression from non-proliferative to proliferative retinopathy in type-2 diabetic patients is a complex multi-mechanism and systemic process (Fig. 4). These proteins may also be potential salivary biomarkers that correlate with progressive stages of diabetic retinopathy. Thus, saliva may be a convenient and less invasive alternative sample to vitreous humor, tear and serum for diabetic retinopathy protein biomarker development.

## ABBREVIATIONS

| | |
|---|---|
| **ACTB** | Actin, cytoplasmic 1 |
| **AGC** | Automatic gain control |
| **AGE** | Glycation end-products |
| **ANXA1** | Annexin A1 |
| **APOA1** | Apolipoprotein A-I |
| **C3** | Complement 3 |
| **CAMP** | Cathelicidin antimicrobial peptide |
| **CAP1** | Adenylyl cyclase-associated protein 1 |

| | |
|---|---|
| **CLU** | Clusterin |
| **ELANE** | Neutrophil elastase |
| **ENO1** | Alpha-enolase isoform 1 |
| **EZR** | Ezrin |
| **FXR/RXR** | Farnesoid X receptor/Retinoid X receptor |
| **GAPDH** | Glyceraldehyde-3-phosphate dehydrogenase isoform 2 |
| **GO** | GO annotation |
| **GSN** | Gelsolin isoform d |
| **HBA1/HBA2** | Hemoglobin subunit alpha |
| **HCD** | Higher-energy collisional dissociation |
| **HP** | Haptoglobin isoform 2 |
| **HSPA1A/HSPA1B** | Heat shock 70 kDa protein 1A/1B |
| **HSPA8** | Heat shock cognate 71 kDa protein isoform 1 |
| **iBRB** | Inner blood retina barrier |
| **IPA** | Ingenuity Pathway Analysis |
| **iTRAQ** | Isobaric tag for relative and absolute quantitation |
| **kV** | kilovolt |
| **LC-MS** | Liquid chromatography-mass spectrometer |
| **LCN1** | Lipocalin-1 isoform 1 |
| **LCN2** | Neutrophil gelatinase-associated lipocalin |
| **LCP1** | Plastin-2 |
| **LDHA** | L-lactate dehydrogenase A chain isoform 3 |
| **LTF** | lactotransferrin isoform 1 |
| **LXR/RXR** | Liver X receptor/Retinoid X receptor |
| **MIF** | Macrophage migration inhibitory factor |
| **MMP9** | Matrix metalloproteinase-9 |
| **MPO** | Myeloperoxidase |
| **MRP8/14** | Myeloid-Related Protein-8/14 |
| **MS** | Mass spectrometry |
| **NPDR** | Type-2 diabetes mellitus with non-proliferative diabetic retinopathy |
| **PDR** | Type-2 diabetes mellitus with proliferative diabetic retinopathy |
| **PKM** | Pyruvate kinase isozymes M1/M2 isoform c |
| **PLTP** | Phospholipid transfer protein isoform a |
| **PRDX1** | Peroxiredoxin-1 |
| **PRTN3** | Profilin-1 |
| **S100A8** | S100 calcium-binding protein A8 |
| **S100A9** | S100 calcium-binding protein A9 |
| **SERPINA1** | Alpha-1-antitrypsin |
| **SIM** | Single ion monitoring |
| **SLPI** | Antileukoproteinase |
| **TIMP1** | Metalloproteinase inhibitor 1 |

| | |
|---|---|
| **TLR-4** | Toll-like receptor-4 |
| **TPM3** | Tropomyosin alpha-3 chain isoform |
| **XDR** | Type-2 diabetes mellitus without diabetic retinopathy. |

## ACKNOWLEDGEMENTS

The authors would like to thank Mr. H. T. Cheah (engineer of Orbitrap Fusion Tribrid Mass Spectrometer) for his technical assistance with mass spectrometer setting, optimization, and data analysis.

### Funding

This work is supported by University of Malaya Research Grant (UMRG) RP006C-13HTM and University of Malaya-Ministry of Education (UM-MoE) High Impact Research (HIR) Grant UM.C/625/1/HIR/MoE/CHAN/13/4 (Account No. H-50001-A000029). The funders had no role in study design, data collection and analysis, decision to publish, or preparation of the manuscript.

### Grant Disclosures

The following grant information was disclosed by the authors:
University of Malaya Research Grant (UMRG): RP006C-13HTM.
University of Malaya-Ministry of Education (UM-MoE) High Impact Research (HIR) Grant: UM.C/625/1/HIR/MoE/CHAN/13/4.

### Competing Interests

The authors declare that they have no competing interests.

### Author Contributions

- Chin Soon Chee conceived and designed the experiments, performed the experiments, analyzed the data, contributed reagents/materials/analysis tools, wrote the paper, prepared figures and/or tables, reviewed drafts of the paper.
- Khai Meng Chang conceived and designed the experiments, performed the experiments, analyzed the data, contributed reagents/materials/analysis tools, wrote the paper, prepared figures and/or tables, reviewed drafts of the paper.
- Mun Fai Loke conceived and designed the experiments, analyzed the data, contributed reagents/materials/analysis tools, wrote the paper, prepared figures and/or tables, reviewed drafts of the paper.
- Voon Pei Angela Loo conceived and designed the experiments, analyzed the data, contributed reagents/materials/analysis tools, wrote the paper, prepared figures and/or tables, reviewed drafts of the paper.
- Visvaraja Subrayan conceived and designed the experiments, analyzed the data, contributed reagents/materials/analysis tools, wrote the paper, prepared figures and/or tables, reviewed drafts of the paper.

## Human Ethics

The following information was supplied relating to ethical approvals (i.e., approving body and any reference numbers):

Medical Ethics Committee of UMMC (Reference number: 1017.28).

## Data Deposition

ProteomeXchange Consortium via the PRIDE:

https://www.ebi.ac.uk/pride/archive/projects/PXD003723;

https://www.ebi.ac.uk/pride/archive/projects/PXD003724;

https://www.ebi.ac.uk/pride/archive/projects/PXD003725.

## Supplemental Information

Supplemental information for this article can be found online at http://dx.doi.org/10.7717/peerj.2022#supplemental-information.

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
