# Peer review of "Association of potential salivary biomarkers with diabetic retinopathy and its severity in type-2 diabetes mellitus: a proteomic analysis by mass spectrometry"

_PeerJ, doi:10.7717/peerj.2022_

## Round 0.1 · original submission · Major Revisions

Dear authors
I agree with the first reviewer in that a major revision is required before the manuscript can be reconsidered.The authors should pay attention to the following points made by the first reviewer.

The authors should validate their findings using e.g. western blot for those most significant proteins.

- Authors refer in the title “orbitrap fusion” and mention superficially the advances in mass spec. however, this is not reflected in the paper. In fact, authors have used iTRAQ for proteome comparison which could be done using other mass spectrometers. So, which is the advantage to use Orbitrap fusion? This issue should be extended or removed and considered a normal proteomic study.
- Why authors have not taken benefit of label-free analysis?
- Authors have not described any validation step. So it is recommended the validation of most significant proteins using a different technique like western-blot.
- A venn diagram will be useful to represent unique proteins instead figure 2.
- Line 358: it will be useful to mention those proteins since are only a few.
Minor points:
Review authors name along text; there are some errors.
Some general reviews concerning saliva and its potential have not been considered.
Authors should also pay attention to the following point made by the second reviewer.
Raw data must be uploaded to a public repository (eg. PRIDE).

Since there is a considerable amount of work to be done, the revision required is a major one.The manuscript should be substantially improved after making the major revision as suggested.

Reviewer 1 ·

Basic reporting

It is an interesting paper that adds knowledge to diabetes type 2 and retinopathy.

Experimental design

The experimental design is adequated to fulfill the main goal. However, more deatil for quantification could be provided.

Validity of the findings

The authors should validate their findings using e.g. western blot for those most significant proteins.

Additional comments

The paper by Chee et al. entitled “Association of potential salivary biomarkers with diabetic retinopathy and its severity in type-2 diabetes mellitus: a proteomic analysis by Orbitrap fusion tribrid mass spectrometer” aims to compare diabetes type 2 with and without retinopathy using saliva. This is an interesting paper and data obtained is of merit to be published since it adds more knowledge to diabetes type 2 and retinopathy. However, there are some points that require the attention of authors before acceptance. These points are:
- Authors refer in the title “orbitrap fusion” and mention superficially the advances in mass spec. however, this is not reflected in the paper. In fact, authors have used iTRAQ for proteome comparison which could be done using other mass spectrometers. So, which is the advantage to use Orbitrap fusion? This issue should be extended or removed and considered a normal proteomic study.
- Why authors have not taken benefit of label-free analysis?
- Authors have not described any validation step. So it is recommended the validation of most significant proteins using a different technique like western-blot.
- A venn diagram will be useful to represent unique proteins instead figure 2.
- Line 358: it will be useful to mention those proteins since are only a few.
Minor points:
Review authors name along text; there are some errors.
Some general reviews concerning saliva and its potential have not been considered.

Reviewer 2 ·

Basic reporting

The manuscript is well written, the text is grammatically correct and easy to understand. Its structure fits to Journal's template, the references are up-date and adequate.
Salivary diagnostic techniques are very important, innovative modern methods for non invasive personal based medical practice.
Figures and figure legends are informative and well designed.
Raw data must be uploaded to a public repository (eg. PRIDE).

Experimental design

Experiments are well design, analytical analyses were carried out by modern, state-of-the-art LC-MS/MS methods.

Validity of the findings

Interpretation and statistical analyses of the results are well done and balanced.

Additional comments

The raw data must be uploaded to a public repository, however the manuscript is suitable for publication.

---

## Round 0.2 · Major Revisions

Please follow the reviewers suggestions to further revise the manuscript.

Reviewer 1 ·

Basic reporting

"No Comments".

Experimental design

"No Comments".

Validity of the findings

I recommend the validation of some proteins that present the elevated fold change by western blot. Specially those which showed really high fold changes.

Additional comments

The authors have introduced some changes in the manuscript. However, the major points remain weakly explained.
For instance, they have referred that saliva has a large dynamic range. To counteract this, iTRAQ should be used. Orbitrap fusion is basically the solution for the analysis. All these concepts were presented without a reference, and it seems to express the author thoughts.

In addition, I have some concerns about the sentence “Table S1 lists the top 26 most up-regulated salivary proteins with a minimum fold change of 20” since it seems to have a higher fold change that I suspect that is outside the iTRAQ dynamic range.

---

## Round 0.3 · accepted · Accept

Your manuscript is now acceptable for publication

Reviewer 1 ·

Basic reporting

no comments

Experimental design

no comments

Validity of the findings

no comments

Additional comments

no comments